# RBAD: A Dataset and Benchmark for Retinal Vessels Branching Angle Detection

Hao Wang[1,†] , Wenhui Zhu[2,†], Jiayou Qin[3,†], Xin Li[2], Oana Dumitrascu[5], Xiwen Chen[1], Peijie Qiu[4], Abolfazl Razi[1,*], and Yalin Wang[2]

[1]School of Computing, Clemson University, Clemson, USA
[2]School of Computing and Augmented Intelligence, Arizona State University, Tempe, USA
[3]Department of Electrical and Computer Engineering, Stevens Institute of Technology, Hoboken, USA
[4]McKeley School of Engineering, Washington University in St. Louis, St. Louis, USA
[5]Department of Neurology, Mayo Clinic, Phoenix, USA

*Abstract*—Detecting retinal image analysis, particularly the geometrical features of branching points, plays an essential role in diagnosing eye diseases. However, existing methods used for this purpose often are coarse-level and lack fine-grained analysis for efficient annotation. To mitigate these issues, this paper proposes a novel method for detecting retinal branching angles using a self-configured image processing technique. Additionally, we offer an open-source annotation tool and a benchmark dataset comprising 40 images annotated with retinal branching angles. Our methodology for retinal branching angle detection and calculation is detailed, followed by a benchmark analysis comparing our method with previous approaches. The results indicate that our method is robust under various conditions with high accuracy and efficiency, which offers a valuable instrument for ophthalmic research and clinical applications. The dataset and source codes are available at: https://github.com/Retinal-Research/RBAD.

*Index Terms*—Medical Imaging, Retinal Analysis, Image Processing, Medical Dataset.

## I. INTRODUCTION

Despite the increasing utilization of deep learning (DL)-based image processing for retinal disease diagnosis, the lack of interpretability and easy annotations remain two significant challenges of such methods, preventing their widespread adoption by medical professionals [1]–[6]. Consequently, many medical practitioners continue to rely on the quantitative analysis of retinal vessel images, as the most prevalent method in modern medicine. Quantitative analysis is indispensable for the early diagnosis, monitoring, treatment planning, and population studies of various retinal diseases. Variations in vascular morphology and structures often indicate disease progression [7], [8]. One notable metric in this context is the **retinal branching angle**, which is instrumental in assessing vascular health and the early detection of systemic diseases such as hypertension and diabetes [9]–[11]. Abnormal branching angles can be indicative of health problems, such as impaired blood flow efficiency due to cardiovascular events.

† Equal contribution
* Corresponding author
This work was partially supported by the grants from NIH (R01EY032125, and R01DE030286), and the State of Arizona via the Arizona Alzheimer Consortium.

These angles are crucial for maintaining the structural integrity of the retinal vasculature [12], with deviations potentially signaling vessel damage. The quantitative analysis of branching angles provides standardized and objective metrics for precise disease diagnosis and monitoring. Additionally, this analysis enhances population studies in retinal research, offering insights into underlying disease mechanisms and supporting the development of innovative diagnostic and therapeutic strategies [13]. By providing a robust framework for evaluating vascular health, the study of branching angles can play a key role in extending our understanding and treatment of retinal and systemic diseases.

Non-invasive retinal imaging is widely used for retinal vascular analysis due to its ability to capture high-resolution and detailed images of the retinal blood vessels. Advances in retinal imaging and its easy accessibility have incubated a series of tools for branching angle detection that assist medical experts in making decisions and streamlining clinical workflow [14], [15]. Specifically, the Retina system (VAMPIRE) [16] and the Singapore 'I' Vessel Assessment program (SIVA) [17] are two popular tools for branching angle analysis. However, these tools are not fully automated and necessitate human experts' intervention (i.e., semi-automated). Recently, there exists an abundant of retinal imaging datasets, exemplified by large-scale datasets like the UK Biobank that consists of more than 100,000 patients [18]. However, these datasets require manual branching angle detection, making them prohibitively impractical to be utilized by medical experts. While the aforementioned semi-automated tools mitigate this issue, conducting branching angle detection for large-scale datasets still remains a time-consuming and labor-intensive task [19], [20]. In addition, existing semi-automated branching angle detection tools are not open-sourced, which hinders the customization of these tools for specialized and streamlined clinical workflows. Therefore, it is inevitable to develop an automated and open-source tool for branching angle detection. Another key challenge in developing branching angle detection tools is the absence of a benchmark dataset, which complicates the evaluation of these tools. Consequently, there is a press-

ing need for to generate a benchmark dataset to assess the performance of branching angle detection tools accurately.

In response to the aforementioned needs, we first introduce an open-sourced annotation tool implemented in Python that can efficiently annotate the branching angle from general retinopathy images. This tool is clinically friendly and can enhance retinal vessel visibility using edge detection and high-pass filters, enabling precise manual annotations. Second, we present a benchmark dataset by applying our customized annotation tool to a popular diabetic retinopathy dataset (i.e., DRIVE dataset [21]) and creating a ready-to-use well-annotated dataset. This dataset comprises 40 retinal images with high-quality annotated branching angles. The annotations are initially conducted by three annotators and further corrected by human medical experts. Subsequently, we propose a bifurcation detection algorithm that efficiently calculates branching angles on retinal segmentation maps.

The contribution of this paper can be summarized as follows: **(i)** We introduce a benchmark dataset for retinal branching angle detection, which is well-annotated by experts with our customized annotation tool, and **(ii)** we proposed an efficient retinal branching angle detection method. We conduct a comprehensive benchmark analysis to demonstrate its superiority by comparing it with recent state-of-the-art algorithms.

In the following sections, we will first introduce the proposed dataset and the annotation tool in Section II. Next, we will provide a detailed description of our branching angle detection and calculation methods in Section III. Subsequently, we will summarize previous work and outline our evaluation metrics, followed by an analysis of the annotated dataset in Section IV. Finally, we will discuss the results of our comprehensive benchmark analysis, highlighting the accuracy and efficiency of our approach compared to existing methods.

## II. BENCHMARK DATASET

### A. Dataset Overview

The DRIVE (Digital Retinal Images for Vessel Extraction) dataset [21] is a key resource in retinal image analysis, supporting the development of algorithms for automatic blood vessel detection. It comprises 40 color fundus photographs, each with a resolution of $565 \times 584$ pixels and a field of view (FOV) of 45 degrees. The dataset is divided into a training set and a validation set, each containing 20 images in JPEG format. 20 manual annotations are provided for each training sample with ground truth segmentations by an expert. The validation set provided by another observer allows for rigorous evaluation of segmentation algorithms. The DRIVE dataset is publicly available from various academic repositories and has become fundamental for retinal disease research, highlighting microvascular complications of diabetic retinopathy. Building on the DRIVE dataset, we present a new dataset that includes 40 retinal vessel images with branching angle annotations. This enhanced dataset aims to provide additional valuable resources for research in retinal image analysis.

### B. Annotation Tool

*1) Interface:* Our annotation tool is designed to promote the fast and accurate annotation of retinal branching angles. The tool is implemented in Python and uses OpenCV library to provide a user-friendly interface. This interface allows users to load retinal images and annotate bifurcation points by simply marking the relevant locations in the image.

*2) Visual Enhancement:* To reduce user fatigue and visual stress during the annotation process, we use different color channels to make retinal vessels and other features more prominent and easier to annotate. Users can dynamically switch between different color channels during annotation, as shown in Figure 1. Our visual enhancement method mainly including:

**Green Channel.** As shown in Figure 1 (c), the green channel reduces the contrast between the background and the vessels while highlighting the vessels. Particularly, it is commonly used as a preprocessing method in unsupervised vessel segmentation algorithms [22], [23]. This selective enhancement allows annotators to focus more easily on the vessels without being distracted by the background.

**Edge Detection.** Figure 1 (d) shows the extracted vessel lines through the Laplacian edge detector [24]. Edge detection algorithms are essential for identifying the boundaries of retinal vessels, making them more distinguishable against the background. The Laplacian edge detector works by calculating the second-order derivatives of the image, highlighting regions of rapid intensity change that correspond to edges. In our implementation, we use a kernel size of 5.

**High-Pass Filter.** Figure 1 (e) applies a high-pass filter to an image using kernel convolution. This process enhances the visibility of high-frequency details by emphasizing edges and fine structures within the retinal vessels [25]. The function accepts three parameters: the image to be processed, the kernel size (51 in our case) for convolution, and an optional boolean to choose between a Gaussian kernel and a mean kernel for blurring. In our implementation, we first calculate the average color value of the image: $\mu = \frac{1}{H \times W} \sum_{i=1}^{H} \sum_{j=1}^{W} I_{norm}(i,j)$, where $I_{norm} = \frac{I}{255}$. Then, depending on the boolean parameter, we apply either a Gaussian blur or a mean blur to the image to obtain the blurred version $I_{\text{blurred}}$. The high-pass filter then subtracts a blurred version of the image (low-frequency components) from the original image: $I_{\text{high-pass}} = I_{\text{norm}} - (I_{\text{blurred}} - \mu)$, thus highlighting the high-frequency components.

Additionally, we implemented useful functions such as mask overlay to separate the annotation from the original image. Users can also edit existing annotations for fast correction.

*3) Instructions:* Users can directly annotate relevant locations within the image display window by selecting three consecutive points, labeled as **a**, **b**, and **c**. Upon selection, these points are automatically connected to form an angle vector, as depicted in Figure **??** (a). This intuitive interface allows users to efficiently identify and mark critical features within the image. The program then employs vector mathematics to

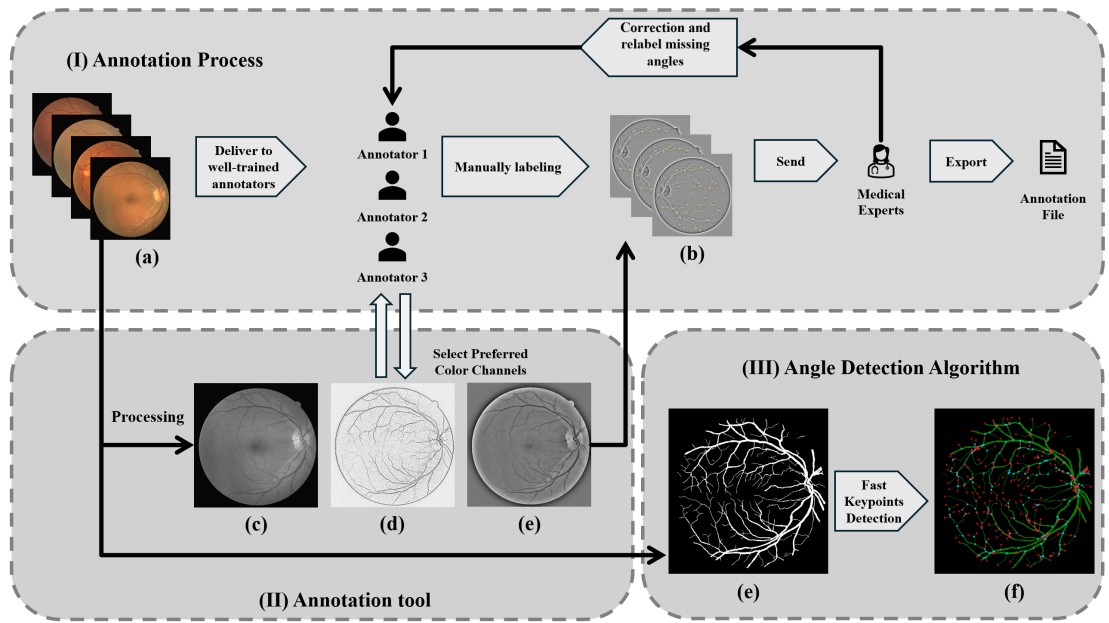

Fig. 1. Framework. (I) shows the annotation process, (b) is the proposed annotation tool, and (c) is the proposed angle detection method. Specifically, (a) is the original RGB image, (b) is the annotated image, (c) is the green channel, (d) is the edge-enhanced image with Laplacian filter, (e) is the image with high-pass-filter, (e) is the segmentation map of RGB image, and (f) is the key points map.

calculate the branching angles formed by the annotated points. This calculation utilizes the coordinates of the three selected points: the bifurcation point and the two branching points. The process is visually represented in Figure **??** (b). The computed angle is immediately displayed on the image, providing users with instant feedback on their annotations. This real-time feedback mechanism enhances the accuracy of the annotation process. If corrections are needed, users can remove the annotated angle by alter-clicking the bifurcation point of the connected vectors, as shown in Figure **??**(c). After the user confirms the annotations, the data, including the coordinates of the points and the calculated angles, are stored in `JSON` format. This structured data format ensures compatibility with other analysis tools, ensuring further processing and analysis. Additionally, annotations can be easily retrieved for subsequent review or integration into larger datasets, promising the safety and accessibility of datasets in future development.

### C. Expert Annotation Protocol

In our study, we employed three annotators to meticulously annotate the blood vessel segmentation images derived from the DRIVE dataset of retinal images. Prior to commencing the annotation work, we provided standardized training to ensure high-quality annotations. This training included detailed instruction on the accurate selection of vessel branching angles in the segmented images, the precision of the annotation angles, and the distinction between true branching angles and artifacts caused by vessel overlap (illustrated in Fig. 3). Moreover, we addressed potential issues arising from sampling instability, such as blurring, residual shadows, and noise introduced by the sampling device itself. To mitigate these issues, we implemented rigorous training sessions aimed at ensuring

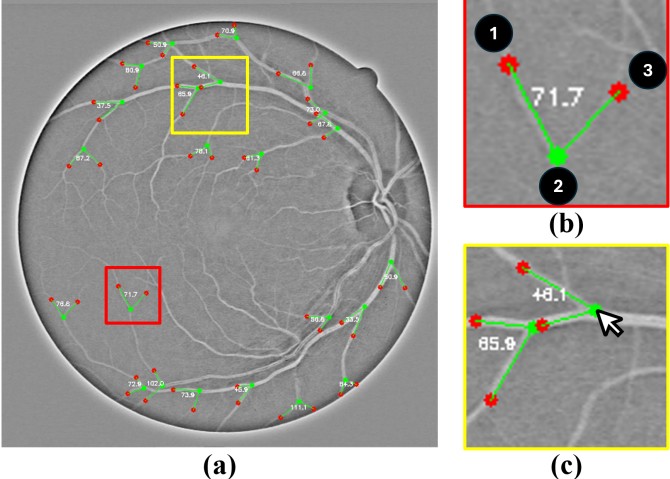

Fig. 2. Interface of Annotation Tool. (a) is the visually enhanced image with angle annotations overlaid. (b) is the annotation process, where an angle annotation is marked with three consecutive clicks, and the second point becomes the bifurcation. (c) shows the annotation editing function, angles can be deleted by alter-clicking (right-click) the bifurcation point.

the validity of the labeled angle information. Each annotator independently annotated all images in the dataset, ensuring thoroughness and accuracy. Following this, we averaged the labeled angles for each retinal image to enhance the stability of the annotations and reduce potential errors. To further ensure that the annotations met medical standards, we submitted the final annotated results for secondary review by a medical professional. Based on the feedback received, the annotators made necessary modifications, which included adding missing annotations and correcting erroneous ones. This rigorous and iterative process ensured the accuracy, standardization, and

medical relevance of each annotated branching angle, thereby enhancing the overall reliability of the dataset.

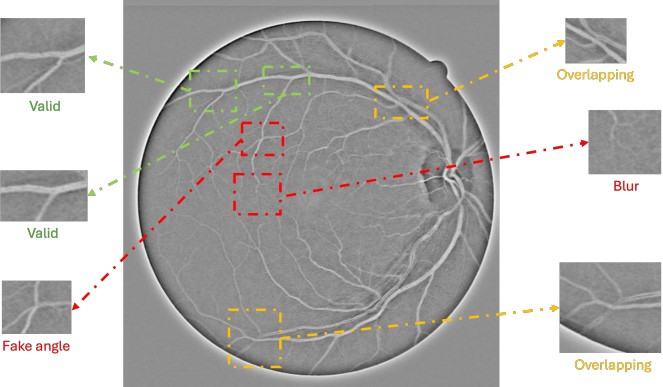

Fig. 3. Some of the problems that tend to occur when labeling are listed, including blurred, overlapping vessels, and false angles due to overlap.

## III. PROPOSED METHOD

As previous key points searching works are designed for general tree-based graphs [26]–[29], we propose a new bifurcation detection and angle calculation method that is specifically designed for retinal images. Our algorithm not only detects bifurcation nodes but also identifies the nearby nodes and their relationships (e.g., child, parent). A bifurcation map generated using our algorithm clearly illustrates the path of detection and calculation for each angle, as depicted in Figure 4 (e). The first step is converting the retinal segmentation map (Figure 4(a)) into a skeletonized representation, where the retinal image is binarized and thinned into 1-pixel-width to highlight the blood vessel centerlines. Then we find a random pixel (non-zero) as the initial point. Assume $I$ represent the skeleton mask:

$$I(x,y) = \begin{cases} 1 & \text{if } (x,y) \text{ is part of the vessel centerline,} \\ 0 & \text{otherwise.} \end{cases} \quad (1)$$

Then, the set of non-zero (vessel centerline) pixels is denoted as:

$$\mathcal{P} = \{(x,y) \mid S(x,y) = 1\}. \quad (2)$$

Then we denote $\tilde{p} \in \mathcal{P}$ as a randomly selected initial point. Subsequently, we apply Fast Key Points Detection to the skeleton mask $I$ by starting from the initial point $\tilde{p}$ to generate the initial key points map. Specifically, we detect potential bifurcation nodes by analyzing pixel connectivity. We use a sliding window approach to traverse the skeleton from the initial point $\tilde{p}$ and examine each pixel within a defined neighborhood to count the number of neighbor pixels. The detailed key points searching process is shown in Figure 4(b) and Algorithm 1.

Subsequently, we apply a Gaussian aggregation to every detected bifurcation to create a heatmap [30], which efficiently identifies the geometric root of the generated tree graph, as shown in Figure 4 (d). Specifically, a Gaussian distribution mask is applied at each bifurcation node $p_k$:

---

**Algorithm 1** Bifurcation Detection
***
**Input:** Skeletonized binary mask $I$
**Output:** Set of key points $\mathcal{K}$
  **Initialization:**
  Define $T = 3$
  Initial key points set $\mathcal{K} = \emptyset$
  Define step count index $c = 0$
  Define node pruning threshold $n = 15$
  Initial point $p_k = \tilde{p} \in \mathcal{P} = \{(x,y) \mid S(x,y) = 1\}$
  Initialize sliding windows $\mathcal{W}$ of size $3 \times 3$ at each seed point
  **while** $\mathcal{P} \neq \emptyset$ **do**
    **for** each active window $W_k \in \mathcal{W}$ centered at $(x_k, y_k)$ **do**
      Calculate nearby pixels for node $k$:

$$N(x_k, y_k) = \sum_{i=-1}^{1} \sum_{j=-1}^{1} I(x_k + i, y_k + j), \quad (3)$$

$$p_k = I(x_k, y_k). \quad (4)$$

      Eliminate visited pixels in $I$:

$$I(x_k, y_k) = 0. \quad (5)$$

      Add 1 on step index $c_k$ for node $p_k$:

$$c_k = c_k + 1. \quad (6)$$

    **if** $N(x_k, y_k) > T$ **then**
      Record $p_k$ as a bifurcation node and add to $\mathcal{K}$ and $\mathcal{P}$
      Generate new windows for each new branch
    **else if** $N(x_k, y_k) \leq 1$ **then**
      **if** $c_k \geq n$ **then**
        Record $p_k$ as a prune node and add to $\mathcal{K}$
      **else**
        Record $p_k$ as an endpoint and add to $\mathcal{K}$
        Erase window $W_k$
      **end if**
    **else**
      Add $p_k$ to $\mathcal{P}$
    **end if**
    **end for**
  **end while**
  **Termination:**
  $\mathcal{P} = \emptyset$
  **return** $\mathcal{K}$

---

$$G_{\text{new}} = \sum G_k(x, y \mid p_k, \sigma_i). \quad (7)$$

Here, $G_k(x, y \mid p_k, \sigma_i)$ represents the Gaussian distribution centered at key point $p_k$ with a standard deviation $\sigma_i$ (typically set to 21 in this study). The resulting image $G_{\text{new}}$ is a heatmap indicating the distribution of bifurcations, where the highest pixel intensity area represents the region with the most gathered bifurcations. As shown in Figure 4 (c), the red circle highlights the region where most bifurcation nodes assemble, identifying it as the root $r$. Thus, we apply our proposed bifurcation detection algorithm again with the new root $r \in \mathcal{P}$ as the start. For each detected bifurcation node, we calculate the angle formed by the branching vessels. This involves computing the vectors of the vessel segments and determining the angle between them. Specifically, we generate

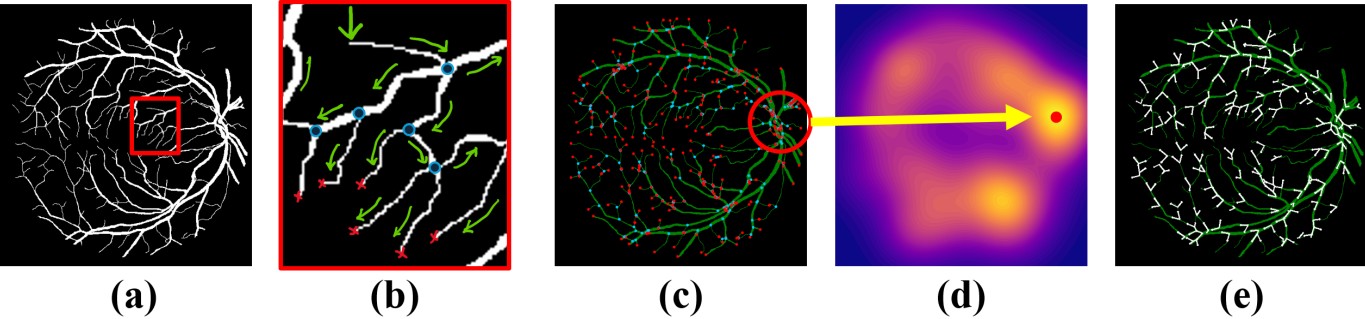

Fig. 4. The process of retina branching angle detection. (a) is the segmentation map, (b) is the Fast Key Points searching process in the selected region of (a), (c) is the key points map, where the red circle indicates an area that has the most bifurcations, (d) is the heatmap of bifurcation points, where the red mark is the location of highest pixel intensity which indicates the root, and (e) is the branching angle map.

a table to describe each key point, as shown in Table I. The branching angle is determined by analyzing the types of the child nodes. For instance, if one child node is a `prune` node and the other is an `endpoint`, they are likely on the same path, indicating an incorrect vector for the branching angle, as shown in Figure 5 (d). Conversely, if both child nodes are `prune` nodes, they are more likely to represent the correct vectors for determining the branching angle as they are located on different routes. The node pruning threshold is set to 15 in our experimental section. Assume $p_k$ is a bifurcation, and $a$ and $c$ are two adjacent points along the vessel segments. The vectors $\vec{v_1}$ and $\vec{v_2}$ are defined as:

$$\vec{v_1} = (a_x - p_x, a_y - p_y),$$
$$\vec{v_2} = (c_x - p_x, c_y - p_y).$$

(8)

The angle $\theta$ between these vectors is given by:

$$\theta = \cos^{-1}\left(\frac{\vec{v_1} \cdot \vec{v_2}}{\|\vec{v_1}\|\|\vec{v_2}\|}\right).$$

(9)

Thus, we record the angle information while the searching window passes through each bifurcation node to generate an angle map, as shown in Figure 4 (e). The proposed random initialization, Gaussian aggregation, bifurcation detection, and angle calculation methods ensure full automation, making it more accessible compared to previous approaches.

TABLE I
THE DESCRIPTION OF ATTRIBUTES OF KEY POINTS WITH THEIR
EXEMPLARY FEASIBLE VALUES.

| Attribute | Definition | Values (Typical/Example) |
|---|---|---|
| index | Index of the key point | 485 |
| (x,y) | Pixel location of the key point | (235, 496) |
| type | Type of key point | root, bifurcation, endpoint, prune |
| step | Pixel distance to its parent | 9 |
| parent | Parent node pixel location | (235, 495) |
| parent_index | Index of the parent node | 484 |
| child | child node | [(235, 497), (236, 496)] |
| child_index | Index of the child nodes | [485,486] |

## IV. BENCHMARK

### A. Previous Work

We summarized and implemented three different methods referenced from previous work that were specifically designed

for retinal bifurcation detection and angle calculation [19], [31], [32]. These methods are:

**Line Detection-based method.** This method detects lines within retinal vessels and finds intersections of these lines as bifurcation points. The angles between the intersecting lines are then calculated as the branching angles, as illustrated in Figure 5 (a).

**ROI-Window-based Detection.** This method calculates branching angles by tracing the vessel branches from the bifurcation point within a region of interest (ROI) window. Initially, the root is identified manually, and a $3 \times 3$ window is used to detect possible bifurcations by checking if the surrounding pixels exceed a certain threshold (e.g., 5-8). For each detected bifurcation, a new window ($50 \times 50$ in our case) is applied to determine the tail of the vectors, and the angle between these vectors is calculated. As depicted in Figure 5 (b), this method provides a structured approach to detecting bifurcation points and calculating angles by focusing on specific regions of interest.

**Rule-based Detection.** This method measures angles by drawing tangents from the bifurcation point along the vessel segments. Acute angles are preferred as branching angles to distinguish actual bifurcations from distorted ones. Pixels with more than three surrounding pixels are identified as bifurcations, and the angles are calculated based on the surrounding sub-pixels. As shown in Figure 5 (c), this method identifies bifurcation points by analyzing pixel connectivity and calculating angles based on the tangents drawn from the bifurcation points.

As illustrated in Figure 5, the Line-Detection-based method **(a)** demonstrates poor detection performance, as the retinal bifurcation branches are not completely straightforward. In contrast, the Rule-Based method **(c)** shows efficient angle detection; however, some angles are misdirected because the method does not account for the relationship with other nodes. The ROI-Window-based method **(b)** produces more robust results compared to (a) and (c), successfully detecting all branching angles with correct directional information. As a comparison, our proposed method (d) surpasses the ROI-Window-based method in angle detection performance. By using the `prune` node as the child of the bifurcation, our method achieves greater precision than approaches relying on

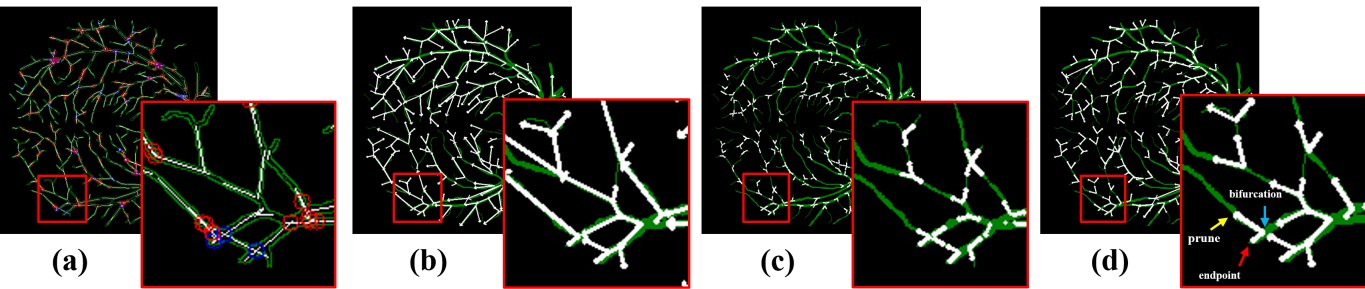

Fig. 5. Retinal angle detection and calculation methods. (a) is the Line Detection-based method, (b) is the ROI-Window method, (c) is the rule-based method, and (d) is our proposed bifurcation detection method with node pruning.

endpoints or other bifurcations as angle vectors.

### B. Evaluation Metrics

To evaluate the effectiveness of these methods for detecting and calculating retinal branching angles, we illustrate the mathematical equations for each metric calculated for each image:

*1) Average Angle:* The average angle for each image is calculated as follows:

$$\bar{\theta} = \frac{1}{n} \sum_{i=1}^{n} \theta_i, \tag{10}$$

where $\bar{\theta}$ is the average angle, $n$ is the number of angle annotations in the image, and $\theta_i$ is the $i$-th angle in the list of angles for that image.

*2) Mean Average Angle :* The mean average angle across all images is calculated as follows:

$$\text{Mean Avg. Angle} = \frac{1}{k} \sum_{j=1}^{k} \bar{\theta}_j, \tag{11}$$

where Mean Avg. Angle is the mean average angle, $k$ is the number of images, and $\bar{\theta}_j$ is the mean angle for the $j$-th image.

*3) Standard Deviation of Angles:* The standard deviation of angles for each image is calculated as follows:

$$\sigma = \sqrt{\frac{1}{n} \sum_{i=1}^{n} (\theta_i - \bar{\theta})^2}, \tag{12}$$

where $\sigma$ is the standard deviation of the angles, $\theta_i$ is the $i$-th angle in the list of angles, $\bar{\theta}$ is the mean angle, and $n$ is the number of angle annotations in the image.

*4) Mean Standard Deviation:* The mean standard deviation across all images is calculated as follows:

$$\text{Mean Std.} = \frac{1}{k} \sum_{j=1}^{k} \sigma_j, \tag{13}$$

where Mean Std. is the mean standard deviation, $k$ is the number of images, and $\sigma_j$ is the standard deviation for the $j$-th image.

*5) Variance of Angles:* The variance of angles for each image is calculated as follows:

$$\sigma^2 = \frac{1}{n} \sum_{i=1}^{n} (\theta_i - \bar{\theta})^2, \tag{14}$$

where $\sigma^2$ is the variance of the angles, $\theta_i$ is the $i$-th angle in the list of angles, $\bar{\theta}$ is the mean angle, and $n$ is the number of angle annotations in the image.

*6) Mean-Variance:* The mean-variance across all images is calculated as follows:

$$\text{Mean Var.} = \frac{1}{k} \sum_{j=1}^{k} \sigma_j^2, \tag{15}$$

where Mean Var. is the average variance, $k$ is the number of images, and $\sigma_j^2$ is the variance for the $j$-th image.

*7) Average Number of Angles:* The average number of angles detected per image is calculated as follows:

$$\text{Avg. Num. of Angles} = \frac{1}{k} \sum_{j=1}^{k} n_j, \tag{16}$$

where Avg. Num. of Angles is the average number of angles, $k$ is the number of images, and $n_j$ is the number of angle annotations in the $j$-th image.

TABLE II
PERFORMANCE BENCHMARK

| Method | Avg. Num. of Angles | Mean Avg. Angle | MAE | MSE | Mean Std. | Mean Var. |
|---|---|---|---|---|---|---|
| Human Annotation | 28 | 71.31 | NA | NA | 18.322 | 348.765 |
| Rule-based | 30 | 83.91 | 12.479 | 185.222 | 23.850 | 591.264 |
| ROI-Window | **134** | 76.71 | 6.724 | 60.582 | 37.686 | 1432.737 |
| Line Detection | 71 | 76.56 | 9.823 | 132.399 | 22.985 | 529.839 |
| **Ours** | 113 | **73.39** | **3.519** | **17.896** | **22.017** | **487.421** |

### C. Data Analysis

We analyzed 40 annotated retinal images, where an average of 28 branching angles are annotated in each image, as shown in Figure 6. Our analysis revealed that the **Mean Average Angle** across all 40 images is 71.31 degrees, and the **Mean Standard Deviation** of the branching angles is 18.32 degrees, suggesting that most branching angles fall within a range of approximately 53 to 89 degrees (71.31 ± 18.32). This value

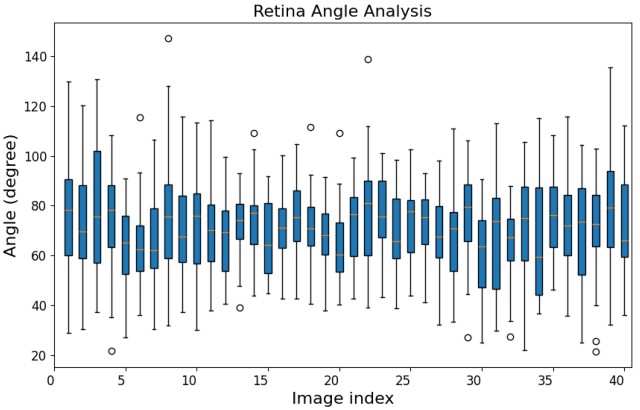

Fig. 6.  Retina Angles for 40 annotated images.

indicates the typical deviation of individual branching angles from the mean, reflecting the variability and dispersion of angles within the dataset. Additionally, the **Mean-Variance** of the branching angles was calculated to be 348.77, which measures the average squared deviation from the mean angle, provides another indicator of the spread of angle values. A higher variance suggests greater variability in the branching angles, while a lower variance indicates more consistency. To further elucidate the distribution of branching angles across our dataset, we visualized the angles for each image, as depicted in Figure 6. This visualization aids in illustrating the range and patterns of branching angles within the data, offering a comprehensive overview of their distribution. This result established a benchmark baseline (**Human Annotation**) for retinal branching angle detection methods. This groundwork is critical for the development and assessment of detection techniques, as the baseline serves as a reference point for evaluating the performance of the implemented detection methods. By providing these standardized metrics, we enable more accurate and reliable comparisons of different detection approaches, thereby advancing the field of retinal branching angle analysis.

### D. Results

We evaluated various methods by inputting the retina images into each algorithm and outputting the predicted retinal branching angles. For each image, we mainly compare the predicted **Mean Average Angle** of the implemented methods with the baseline using **Mean Absolute Error** (MAE) and **Mean Squared Error** (MSE). As shown in Table II, the ROI Window-based method previously achieved the best results among earlier works, detecting an average of 134 angles with a mean average angle of 76.71 degrees, an MAE of 6.724, and an MSE of 60.582. However, our proposed method surpasses all prior approaches. Specifically, our method detects an average of 113 angles with a mean average angle of 73.39 degrees, an MAE of 3.519, and an MSE of 17.896. To put these results into context, when the average angle for a retinal image is 71 degrees, our predicted value is 73 degrees. This is significantly more precise compared to the Rule-based method, which had an MAE of 12.479 and an MSE of 185.222, and

the Line method, which had an MAE of 9.823 and an MSE of 132.399. While the ROI Window method detected more angles on average (134 compared to 113), our method significantly outperforms it in terms of accuracy. Our proposed method achieves a much lower MAE of 3.519 compared to the ROI Window's MAE of 6.724, demonstrating more precise angle predictions. Additionally, our method maintains a competitive **Mean Standard Deviation** of 22.017, and **Mean-Variance** of 487.421, which is very close to the **Human Annotation** (18.322/348.765), even with approximately **four** times the number of detected angles. This indicates that our predictions are not only more accurate but also more consistent with the actual distribution of branching angles in the retinal images. Therefore, the performance of our method demonstrates its robustness and reliability in detecting retinal branching angles, providing precise angle predictions from the observed retinal images.

### V. Discussion

Our study offers several contributions to the detection and annotation of retinal branching angles, yet there are areas where further improvements can enhance the methodology and its applications. Firstly, the implemented ROI-Window-based method demonstrates significant potential for improvement, given its structural similarity to our proposed method. This method already shows robust performance in detecting branching angles with accurate directional information, which can be further improved by incorporating our pruning method, as our node pruning can reduce outliers by tuning its threshold. Secondly, while our method for finding the root is highly efficient, its robustness can be further enhanced. Some retinal images may exhibit xenomorphic characteristics, presenting challenges for accurate root identification. Future work should focus on improving the algorithm's ability to handle such variations. Thirdly, our annotation tool, which facilitates the accurate marking of branching angles, can be further improved by integrating it with our proposed angle detection method. This integration would enable users to quickly generate an estimated angle map with a single click, significantly speeding up the annotation process. Users could then fine-tune these preliminary annotations, combining the efficiency of automated detection with the precision of manual correction. This enhancement would make the annotation process more efficient and user-friendly, particularly for large datasets. Lastly, in our ongoing efforts to enhance the usability of our annotation tool, we are committed to making it more accessible and clinic-friendly for a broader range of users including those with limited technical expertise to boost the medical research community.

### VI. Conclusion

In this study, we introduced an annotation tool that enhances retinal vessel visibility through color channels and edge detection, improving annotation accuracy and efficiency. We also created a dataset of 40 high-quality retinal fundus images with branching angle annotations, validated by professional annotators and medical doctors. This dataset fills the gap in

high-quality resources for retinal branching angle analysis. Our method for automatic detection of vascular branching angles outperforms previous approaches in terms of MAE and MSE, with highly consistent results. We offer open-source algorithms and datasets to extend retinal analysis and support more accurate diagnoses, contributing to medical imaging and ophthalmology.

## ACKNOWLEDGMENT

This work was partially supported by the grants from NIH (R01EY032125, and R01DE030286), and the State of Arizona via the Arizona Alzheimer Consortium.

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
