# OpenReview forum: "RBAD: A Dataset and Benchmark for Retinal Bifurcation Angle Detection"
_IEEE.org/EMBS/BHI/2024/Conference — IEEE BHI'24_

### Official Review · Reviewer_TE1C · 2024-08-07
**Innovative approach and high potential for impact**

**Overall Rating:** 8
**Confidence:** 5

**Other Quality Metrics:**

•	Clarity of writing: great
•	Clinical significance: excellent
•	Methodological novelty: excellent
•	Experiments and results: excellent

**Questions For The Authors:**

None

**Strengths:**

The method demonstrates high accuracy and efficiency in comparison to previous approaches. The authors emphasize the importance of the open-source tool and dataset for facilitating further research in retinal image analysis. The work addresses a significant clinical problem, has an Innovative approach and high potential for impact.

**Summary Of The Paper:**

The paper addresses the challenge of accurately and efficiently detecting retinal bifurcation angles, a crucial metric for diagnosing various eye diseases. To overcome limitations of existing methods, the authors developed an open-source annotation tool and a benchmark dataset consisting of 40 retinal images with annotated bifurcation angles. Additionally, they propose a new method for automatically detecting and calculating these angles.

**Weaknesses:**

The method demonstrated a highly reliable performance. While its ability to identify variations or outliers is somewhat limited, this is an area for future development. Furthermore, making the method more accessible would be beneficial

---

### Official Review · Reviewer_GeAs · 2024-08-10
**Review of RBAD**

**Overall Rating:** 7
**Confidence:** 4

**Other Quality Metrics:**

(a) Clarity of writing: good
(b) Clinical Significance: perfect
(c) Methodological Novelty: fair
(d) Experiments and Results: good

**Questions For The Authors:**

1. What is the difference in automation between this paper and other work (semi-automated)? Can be more detailed?
2. Why does the evaluating metirc contains the average number of angles? Do evaluating metrics used in object detection (like sensitivity, specificity, et al) are more suitable?

**Strengths:**

The paper presents an open-source annotation tool, which has clinical significance. Based on the tool, the authors present a dataset,which can used by other researchers for thier research. The description of the algorithm is detailed and clear. It provides a new idea for retianl bifurcation angle detection.

**Summary Of The Paper:**

This work contains the following main contents: (1)introduce an open-source annotation tool implemented in Python; (2) present a benchmark dataset by applying our customized annotation tool to a popular diabetic retinopathy dataset; (3) propose a bifurcation detection algorithm that efficiently calculates bifurcation angles on retinal segmentation maps

**Weaknesses:**

1. There are some clerical errors in the text
2. The explaination of prune node and endpoint may be insufficient. It is better if it can be illustrated by figure.

---

### Decision · Program_Chairs · 2024-09-23

Accept